# Living with "long COVID": A systematic review and meta-synthesis of qualitative evidence

M. Mahbub Hossain[1,2]*, Jyoti Das[3], Farzana Rahman[4], Fazilatun Nesa[3], Puspita Hossain[5,6], A. M. Khairul Islam[7], Samia Tasnim[8], Farah Faizah[9], Hoimonty Mazumder[9], Neetu Purohit[10], Gilbert Ramirez[11]

1 Department of Decision and Information Sciences, C.T. Bauer College of Business, University of Houston, Houston, Texas, United States of America, 2 Department of Health Systems and Population Health Sciences, Tilman J. Fertitta Family College of Medicine, University of Houston, Houston, Texas, United States of America, 3 Research Initiative for Health Equity, Khulna, Bangladesh, 4 Armed Forces Medical College, Dhaka, Bangladesh, 5 McMaster University, Hamilton, Ontario, Canada, 6 BRAC James P Grant School of Public Health, BRAC University, Dhaka, Bangladesh, 7 International Centre for Diarrhoeal Disease Research, Bangladesh (icddr,b), Mohakhali, Dhaka, Bangladesh, 8 School of Public Health, Texas A&M University, College Station, Texas, United States of America, 9 School of Public Health, University of Memphis, Memphis, Tennessee, United States of America, 10 The IIHMR University, Jaipur, Rajasthan, India, 11 Robert Stempel College of Public Health and Social Work, Florida International University, Miami, Florida, United States of America

* mhossa25@central.uh.edu

**Data Availability Statement:** All data associated with this manuscript are retrieved from published materials available in the referred journal articles.

## Abstract

### Objectives

Long-term health consequences of coronavirus disease (COVID-19), also known as "long COVID," has become a global health concern. In this systematic review, we aimed to synthesize the qualitative evidence on lived experiences of people living with long COVID that may inform health policymaking and practice.

### Methods

We searched six major databases and additional sources and systematically retrieved relevant qualitative studies and conducted a meta-synthesis of key findings using the Joanna Briggs Institute (JBI) guidelines and reporting standards of the Preferred Reporting Items for Systematic Reviews and Meta-Analysis (PRISMA) checklist.

### Results

We found 15 articles representing 12 studies out of 619 citations from different sources. These studies provided 133 findings that were categorized into 55 categories. All categories were aggregated to the following synthesized findings: living with complex physical health problems, psychosocial crises of long COVID, slow recovery and rehabilitation, digital resources and information management, changes in social support, and experiences with healthcare providers, services, and systems. Ten studies were from the UK, and others were from Denmark and Italy, which highlights a critical lack of evidence from other countries.

**Funding:** The author(s) received no specific funding for this work.

**Competing interests:** The authors have declared that no competing interests exist.

## Conclusions

More representative research is needed to understand long COVID-related experiences from diverse communities and populations. The available evidence informs a high burden of biopsychosocial challenges among people with long COVID that would require multilevel interventions such as strengthening health and social policies and services, engaging patients and caregivers in making decisions and developing resources, and addressing health and socioeconomic disparities associated with long COVID through evidence-based practice.

## Background

The coronavirus disease (COVID-19) pandemic has resulted in a growing burden of biopsychosocial problems globally [1]. A growing body of knowledge about COVID-19 informs how SARS CoV-2, the virus causing COVID-19, spreads quickly among individuals and populations, leading to high mortality and morbidity [2–4]. The acute nature of the COVID-19 symptoms necessitated global attention to understand how it spreads, manifests, how to prevent it in populations, and how to treat it in clinical settings [5–7]. Over time, another phenomenon of this pandemic emerged in scientific discourses that emphasized the long-term consequences of COVID-19 that persist beyond the acute phase of illness [8–10]. With varying definitions and timelines following the primary infection, such long-term health problems are commonly termed as "long COVID," "post COVID," "post-acute COVID," "post-acute sequelae of SARS CoV-2 infection (PASC)," "chronic COVID," and "long-haul COVID [8,11–13]".

Emerging scientific literature recognizes a heavy burden of long COVID conditions such as fatigue, breathlessness, post-exertional malaise, cough, chest pain, headache, sleep problems, difficulties in thinking (also known as "brain fog"), lightheadedness, musculoskeletal pain, depression, anxiety, and other health problems [5,10,12,13]. A growing body of research discusses the nature and magnitude of long COVID that may affect different organs and body systems among the affected individuals [14]. Studies have reported cardio-respiratory symptoms, gastrointestinal problems, musculoskeletal disorders, and neuro-cognitive imparments alongside generalized health problems attributable to disorders of multiple systems [14–17]. A living systematic review of 39 studies found a high prevalence of common weakness (41%; 95% confidence interval [CI]: 25–59%), poor quality of life (37%; 95% CI: 18–60%), generalized malaise (33%; 95% CI: 15–57%), fatigue (31%; 95% CI: 24–39%), concentration impairment (26%; 95% CI: 21–32%) and breathlessness (25%; 95% CI: 18–34%)and amongst common characteristics of long COVID [17]. Most people with acute symptoms of COVID-19 may recover within a few days to a few weeks, whereas many individuals experience long COVID symptoms after four weeks or later of being infected with SARS CoV-2 [8,10,12]. A longitudinal study of 170 individuals from Denmark, who were previously diagnosed with COVID-19, reported that 39% of the participants experienced persistent symptoms (median 168 [93–231] days) after the acute phase and 8% (95% CI: 5–13%) reported severe persistent symptoms [18]. Another cohort study of 118 Moroccan individuals found that the prevalence of long COVID symptoms was 47.4%, with a significantly high burden of asthenia, myalgia, and brain fog [19]. These findings suggest a high burden of long COVID related problems across populations.

A growing body of evidence from many observational studies globally has been synthesized in recent meta-research. A systematic review identified 37 articles and found health problems

such as fatigue (16–64%), dyspnea (15–61%), cough (2–59%), arthralgia (8–55%), and thoracic pain (5–62%) that were prevalent among people with long COVID [20]. A meta-analysis of 41 studies reported the overall global prevalence of long COVID conditions was 43% (95% CI: 39%-46%), whereas the prevalence after 30, 60, 90, and 120 days following the infection were estimated to be 37% (95% CI: 26%–49%), 25% (95% CI: 15%–38%), 32% (95% CI: 14%–57%), and 49% (95% CI: 40%–59%), respectively [12]. These studies inform a high burden of long COVID conditions with marked variations in symptoms and durations that require an in-depth understanding of how these long-term health problems affect individuals.

Qualitative insights into their physical, mental, social, and healthcare experiences may complement the quantitative evidence on long COVID [21–23]. Moreover, synthesized qualitative evidence may enable healthcare researchers, professionals, and policymakers to better understand the health and social needs of people with long COVID and address the same with evidence-based decision-making and practices [21,24]. This systematic review and meta-synthesis aimed to identify and synthesize qualitative evidence on the experiences of people with long COVID to inform clinical and public health decision-making that may alleviate health disparities among the affected individuals and populations.

## Materials and methods

### Methodological guidelines

We conducted this systematic review adhering to the Joanna Briggs Institute (JBI) methodology for qualitative systematic review and reported in accordance with the Preferred Reporting Items for Systematic Reviews and Meta-Analysis (PRISMA) guidelines [25,26]. The completed PRISMA checklist is available in S1 File. The protocol of this review was prepared in priori that was not registered with any review registries. The complete protocol for this systematic review is available upon request.

### Data sources and search strategy

We systematically searched Medline, Web of Science, Academic Search Ultimate, Health Policy Reference Center, American Psychological Association (APA) PsycInfo, and the Cumulative Index to Nursing and Allied Health Literature (CINAHL) databases using a set of keywords with Boolean operators. Also, we performed a manual search in Google Scholar and examined additional sources such as reference list assessment, consultation with subject matter experts, and citations analyses of previous research on long COVID. No language or geographic restrictions were applied during the search process. The preliminary search was performed on July 1, 2020, and the search was updated on June 6, 2022 (Table 1).

### Eligibility and screening of the literature

We used a set of eligibility criteria to identify and include individual studies in this review. An article was considered eligible if it:

a. included people with long COVID irrespective of how long COVID was defined or measured in their respective contexts,

b. reported the experiences, perspectives, or opinions of the participants on any aspect of living with long COVID conditions,

c. used a qualitative methodology or mixed methods with qualitative components with any analytic approach (e.g., thematic analysis, ethnography, action research etc.),

**Table 1. Search strategy for systematic literature retrieval.**

| Search step | Keywords applied in titles, abstracts, topics, and subject-specific headings |
|---|---|
| 1 | "Coronavirus" OR "COVID*" OR "COVID-19" OR "COVID19" OR "SARS-CoV-2" OR "2019 novel coronavirus" OR "2019-nCoV" |
| 2 | "Long COVID*" OR "long haul*" OR "long-haul*" OR "post-acute COVID*" OR "late sequela COVID*" OR "persistent COVID*" OR "chronic COVID*" OR "long term COVID*" OR "long-term consequence*" OR "long-term impact*" OR "long-term effect*" OR "post-acute" OR "long-tail" OR "persist* symptom*" OR "prolonged symptom" |
| 3 | "qualitative stud*" OR "qualitative research" OR "qualitative evidence" OR "qualitative*" OR "thematic analysis" OR "grounded theory" OR "focus group" OR "participant observation" OR "phenomenology" OR "critical theor*" OR "interpretative" OR "ethnograph*" OR "action research" OR "mixed method*" |
| Final search strategy | 1 AND 2 AND 3 |

d. was published as an original article in a peer-reviewed journal, and

e. had the full text available in the English language.

These criteria were set upon a series of discussions among the authors and subject matter experts, emphasizing the inclusiveness of qualitative evidence from a diverse body of scientific literature. For example, all definitions of long COVID or all qualitative methods were acceptable as the transdisciplinary scholarship may have high methodologic heterogeneity. Moreover, restrictions on publication type ensured the inclusion of original studies; therefore, other publications such as reviews, commentaries, editorials etc. were excluded from this review.

We screened all citations retrieved from the search strategy using a cloud-based software (rayyan.ai). Two authors (AD and FR) independently screened each citation using the eligibility criteria, and all conflicts after the first round of review were addressed through discussion with a third reviewer (MMH). This process was repeated for full-text evaluation, and all articles eligible after this stage were used for data extraction and analysis.

## Data extraction

We used a data extraction tool that included items such as study descriptions, place and timeline of the respective study, sample characteristics, study methods and materials, key findings, and conclusions of the authors. Moreover, we extracted study findings as direct statements and metaphors, ensuring the representation of the authors' works in the respective study. In this process, each finding was accompanied by one or more illustrations that were extracted as quotes of the study participants. According to the JBI methodology, each finding was assigned a level of credibility. Findings that were supported by illustrations demonstrating evidence beyond any reasonable doubt were considered "unequivocal," whereas findings with illustrations that could be challenged due to lack of data were considered "credible." Lastly, findings that were not supported by any illustrations from the participants were considered "unsupported."

## Quality assessment

We used the JBI Critical Appraisal Checklist for Qualitative Research Synthesis to evaluate the methodological quality of the included studies. Two reviewers (AD and FN) independently performed the quality evaluation, and all disagreements were resolved through a discussion with another reviewer (MMH).

## Data analysis and meta-synthesis

We synthesized the qualitative data using the JBI meta-aggregation approach that aimed to explore generalizable statements in the form of evidence-based recommendations for health-care decision-makers and practitioners. In this approach, the synthesized evidence remains sensitive to the interpretation of the primary authors that is both practical and usable in respective contexts. All unequivocal and credible findings extracted from primary studies were grouped to categories based on similarities in their meanings. Further, multiple categories with similar themes were grouped to the synthesized findings. All findings, categories, and synthesized findings were reviewed and discussed by the review members to reach a consensus on the meta-aggregation of qualitative evidence.

## Results

### Literature retrieval

As depicted in Fig 1, we identified a total of 619 citations from the databases and additional sources. After eliminating the duplicates, we reviewed the titles and abstracts of 262 unique citations and found 36 articles eligible for review. Further, we retrieved and examined the full texts of those citations and excluded 21 that did not meet all criteria. Finally, 15 articles representing 12 studies were included in this qualitative systematic review and summarized in Table 2 [27–41].

### Characteristics of the included studies

Most studies (n = 10, representing 13 articles) were conducted in the UK, and one study each was conducted in Denmark [33] and Italy [32]. Three studies collected data until 2021, whereas most (n = 8) studies completed data collection in 2020. Most (n = 11) studies used thematic analysis; only one study used a phenomenological approach for qualitative analysis. Digital media such as email, phone, and teleconferencing technologies were used for data collection across the included studies. The sample sizes ranged from 11 to 3290, with a total sample size of 3849 in 12 studies. In 12 out of 15 studies, female were higher in proportion (up to 80%) among the participants from all population sub-groups. Most studies reported adult participants with age ranging from 18 to 82 years. Patient support groups were the major sources of recruiting participants across the included studies.

### Quality appraisal

Twelve out of fifteen articles had eight or more positive responses in ten items quality appraisal checklist, which suggests a low risk of bias across those studies. Three articles had six to seven positive responses that suggest moderate risk of bias, whereas no articles were found to have less than six positive responses. None of the included articles provided any evidence on locating the researchers within the cultural context of respective studies or described the influence of the researchers on research and vice versa. The summary of the quality appraisal is available in the S2 File.

### Qualitative meta-synthesis

A total of 133 findings were identified from 12 studies that were categorized into 55 categories. These categories were further classified within six synthesized findings.

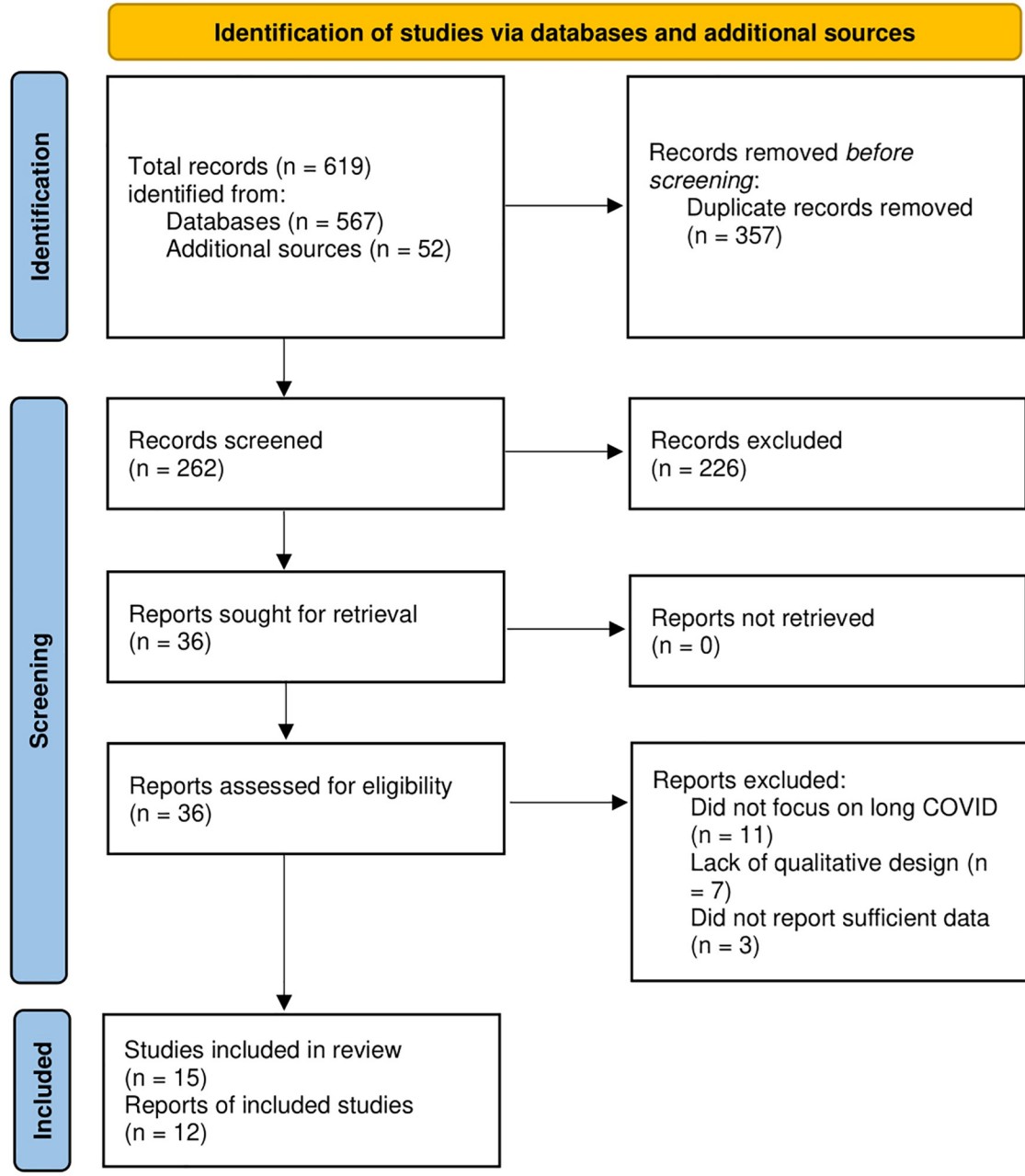

**Fig 1. Flow diagram of the literature review process.**

### Synthesized finding 1: Living with complex physical health problems

People living with long COVID reported complex and chronic physical health problems that were identified in 37 findings (16 unequivocal and 21 credible) in 12 categories that are summarized in Table 3. Participants reported major challenges associated with persistent symptoms such as breathlessness [27,32,34,38] and fatigue that affected their wellbeing.

**Table 2. Characteristics and key findings of the included studies.**

| Authors; publication year | Study location and time | Study design and analysis | Data collection procedure and tools | Sample size; Characteristics of the participants; recruitment strategy | Key findings and authors conclusions |
|---|---|---|---|---|---|
| Ladds et al. (2020) [28] | UK; May-September, 2020 | Qualitative; constant comparative method | Narrative interview (n = 55), focused group discussion (n = 8) using telephone, video call, email | 114; Female (n = 80), male (n = 34) doctors (n = 32), other health professionals (n = 19), general patients (n = 63), attended hospital (n = 31), age range (27–73 years); purposive sampling followed by snowballing via covid-19 patient support groups, social media | Patients showed diverse illness with uncertain treatment standard and prognosis causing deteriorated outcome. Patients felt to be stigmatized, having difficulty in achieving diagnosis due to lack of accessing service, disjointed and siloed care as well as biased therapeutic care service. It is suggested to develop quality principles for long covid-19 patients to ensure accessing services, rendering continuity of evidence-based care followed by multidisciplinary rehabilitation and further advancement of service. |
| Kingstone et al. (2020) [27] | UK; July-August, 2020 | Qualitative; thematic analysis using constant comparison method | Semi- structured interview (n = 24) using telephone, video call, email | 24; Female (n = 19), Male (n = 5), age range (20–68 years); Purposive sampling with snowballing from social media, peer support group | The article revealed the patients faced challenges in managing their symptoms and accessing care besides the feeling of uncertainty with helplessness. This will also help to raise awareness among primary care professionals regarding the long covid symptoms and the importance of expressing their empathy and support during the recovery and rehabilitation. |
| Buttery et al. (2020) [34] | UK; April-December, 2020 | Mixed method study; thematic analysis | Online survey | 3290; Female (78%), age range (45–54 years), Hospitalized: 17.7%; NR | The qualitative analysis expressed patient's experience of unpredictable physical and psychological symptoms of covid-19 with the unsatisfactory healthcare service. And the evaluation of their experience, society, healthcare system will assist to roadmap future service planning to address these symptoms persistence problem. |
| Høier et al. (2021) [33] | Denmark; NR | Qualitative; thematic analysis | Semi- structured in-depth interview (n = 19) using telephone, video call | 19; Female (n = 17), male (n = 2), age range (25–66 years); purposive sampling using social media support groups | The study focused on themes like well-functioning sense and flavor of food, eating environment and post-eating well-being. As covid-19 causes the chemosensory dysfunction, shifting focus towards choosing healthy food and texture, trigeminal stimulation, and joyful eating environment can boost well-being around food intake. This should be incorporated into health promotion initiatives for the recovering patients. |
| Humphreys et al. (2021) [35] | UK; September-October, 2020 | Qualitative; reflexive thematic analysis | Semi- structured interview (n = 19) using telephone | 18; Female (n = 9), male (n = 9), age range (18–74 years); purposive sampling from covid-19 research database | The findings are categorized in themes such as, participant's struggle to cope up with reduced physical activity, self-acceptance of this reduction as well as the fear of this to be permanent, challenges in finding appropriate physical activity advice, approaches to manage symptoms like fatigue, brain fog while trying to resume daily activities and exercise. So, well-tailored advice and support to recommence physical activity for long haulers are recommended. |

*(Continued)*

**Table 2.** (Continued)

| Authors; publication year | Study location and time | Study design and analysis | Data collection procedure and tools | Sample size; Characteristics of the participants; recruitment strategy | Key findings and authors conclusions |
|---|---|---|---|---|---|
| **Ladds et al. (2021)** [36] | UK; May-September, 2020 | Qualitative; thematic analysis | Narrative interview, focused group discussion using email, phone call and video recording, contemporaneous notes | 43; Female (n = 35), male (n = 8), age range (32–58 years), hospitalized (n = 7); purposive sampling followed by snowballing from online COVID-19 support group, social media | The article has demonstrated a set of co-developed multidisciplinary evidence based standard approaches which ensure equitable and accessible care service with less patient burden and more responsibility and patient involvement. This can be practiced through a potential care pathway model to improve the management of long covid patients. |
| **Razai et al. (2021)** [37] | UK; November, 2020 | Qualitative; thematic analysis | Interview using semi-structured questionnaire via telephone | 41; Female (n = 27), male (n = 14, age range (19–82 years), worse QoL: 14, accessed to GP: 30; Simple random sampling from two primary care services | The most common symptoms found are fatigue (45%), breathlessness (30%), neurocognitive problem (30%). Patients also experienced little information on safety advice and feeling of fear with isolation. Half of the patients (56%) expected the healthcare staffs to regular follow-up them, and later this has been adapted by the care practices where social prescribers monitored the long covid patients and referred to the GPs, nurses, community support group accordingly. |
| **Shelley et al. (2021)** [38] | UK; NR | Qualitative; inductive thematic analysis | Semi-structured interview via Video conferencing software | 48; Female (n = 41), male (n = 7), age range (47+/- 7 years); purposive sampling from social media and university webpage | The article categorized the experience of patients in themes such as, daily life with managing COVID-19, self-management techniques, restoration of regular physical activity, and challenges to go back to works. This resume of activities often causes the symptoms worse so that customized strategies should be planned to support these patients dealing with difficulties to recover. |
| **Taylor et al. (2021)** [39] | UK; July-August, 2020 | Qualitative; inductive thematic analysis | Semi-structured interview via telephone, video call | 13; Female (n = 11), male (n = 2), age range (30–50 years); Purposive sampling followed by snowballing from social media and peer support group using email | The findings of this article fit in different themes like experiencing covid-19 symptoms, falling short of expectation, using technical knowledge and communication, helping attitude and expressing identity. Participants seemed to be both hopeful for good and distressful for being let down. However, they agreed that they take better care of patients with chronic condition. |
| **Burton et al. (2022)** [40] | UK; November, 2020-September, 2021 | Qualitative; reflexive thematic analysis | Semi-structured interview via telephone (n = 7), video (n = 14) | 21; Female (n = 14). male (n = 7), age range (26–70 years); Purposive sampling from online support group, university newsletter advertisement, social media | Five themes were developed according to participant's experiences which are symptoms disrupting daily life, limitations of treatment options, uncertain course of sickness, experience regarding care and the changed identity. These negative factors can be mitigated by adopting patient centric health service and more research on peer support group, self-management approach as well. |

*(Continued)*

**Table 2.** (Continued)

| Authors; publication year | Study location and time | Study design and analysis | Data collection procedure and tools | Sample size; Characteristics of the participants; recruitment strategy | Key findings and authors conclusions |
|---|---|---|---|---|---|
| **Callan et al. (2022)** [41] | UK; October- November, 2020 | Qualitative; thematic analysis using constant comparison method | Online interview (n = 23), focused group discussion (n = 5); email | 50; Female 80%; purposive sampling with snowballing from a previous study sample (n = 23), online support group (n = 27) | The findings are categorized in different themes, such as, mixed opinion towards the word "brain fog", lived neurocognitive difficulties, fluctuating sickness, mental impact on personal to professional relations and recognition, feeling of guilt, shame, stigma, self-management approaches, challenges in attending healthcare service and describing symptoms. An ongoing therapeutic relationship has been recommended to develop to address these issues. |
| **Day (2022)** [29] | UK; July, 2021 | Qualitative; thematic analysis | Online semi-structured interview; video call | 11; Female (n = 8), male (n = 3), age range (20–59 years); convenience sampling from online support groups, social media | The role of online peer support groups can be expressed under 5 themes, such as feeling up gaps in service, community awareness, engagement, diversity, and social communication. However, these groups can be both beneficial and harmful for improvements. So, by careful use and prioritizing patient concern, the benefits of this groups can be facilitated. |
| **Ireson et al. (2022)** [30] | UK; April- September, 2020 | Qualitative; thematic analysis | Online submitted stories | 66; NR; NR | The highlighting themes from the study are difficulties with impact of the illness, regarding validation and even choosing alternative management option. Long covid treatment requires objective indicators rather than following evidence-based practice and patient testimonies should be given importance to remove stigma and for better therapeutic relation. |
| **Khatri (2022)** [31] | UK; November 2020- February, 2021 | Qualitative; thematic analysis | Focus group discussion (n = 5), using email. | 47; Female (n = 20), male (n = 27), age range (18–70 years); Convenience sampling | According to the findings of the study, altered taste and odor in food consumption affects the nutritional food value, healthy weight, type of food, anxiety, and social interconnection. As olfactory function has important role on food intake with the related quality of life, early intervention should be taken to mitigate the condition. The findings also help medical practitioner to understand the food consumption pattern of the patients. |
| **Schiavi et al. (2022)** [32] | Italy; June- July, 2020 | Qualitative; empirical phenomenological approach | Interview using telephone, previous hospital record check | 150; Female (n = 94), male (n = 56), age range (62.8 +/ 11.8 years); NR | The experiences of the participants are included in themes like continuing symptoms, fear, stigmatized isolation feelings, fatalistic attitude, and restoration of daily life. |

(Abbreviation: NR—Not reported).

**Table 3. Findings and categories within synthesized finding 1.**

| Synthesized finding 1 | Categories | Findings |
|---|---|---|
| **Living with complex physical health problems** | Breathlessness and fatigue | Fatigue and breathlessness (U) |
| | | Ongoing fatigue(U) |
| | | Breathlessness affecting regular conversations (U) |
| | Difficulties in performing daily activities | Limitations on daily activities (U) |
| | | Difficulties with performing leisure activities (C) |
| | | Restricted abilities to perform normal activities (U) |
| | | Persistent symptoms affecting daily activities (C) |
| | Persistent symptoms | Experiencing and enduring persistent symptoms (U) |
| | | Interconnection of physical and psychological symptoms(C) |
| | | Making sense of symptoms © |
| | | Rollercoaster symptoms affecting different body systems in different times (C) |
| | Living in uncertainty | Uncertain and confusing illness (U) |
| | | Fear, uncertainty, and despair (U) |
| | | Uncertainty regarding symptoms and prognosis (U) |
| | | Fear, Isolation, and Uncertainty Related to COVID-19 (U) |
| | Prolonged impacts on patients' lives | Impact of long COVID on patients' lives (C) |
| | | Re-initiating physical activity (C) |
| | | Less control over life (C) |
| | Coping with the new lifestyle | Adapting to an altered life (C) |
| | | Learning to adapt (U) |
| | | Returning to adapted life course (C) |
| | Cognitive impairments | Decline in neurocognitive functions(C) |
| | Variations in recovery | Different stages of recovery (C) |
| | Relapse of health problems | Relapsing symptoms(U) |
| | Concerns about recovery | Concerns regarding recovery (C) |
| | Changes in taste and smell | Effects on appetite (C) |
| | | Odor perception (C) |
| | | Prolonged loss or changes in taste or smell (C) |
| | | Altered taste perception (C) |
| | | Changes in smell associated with temperature of the food (C) |
| | | Coping with altered taste and smell (C) |
| | Food consumption behavior | Altered food-related sensory perceptions (U) |
| | | Chemesthesis perception (C) |
| | | Food-related pleasure (U) |
| | | Coping with changes in appetite (U) |
| | | Focus on familiar food items (C) |
| | | Eating environment (U) |

*"When I take a long walk, I become short of breath"*

*([32], p4)*

*"If I run upstairs really quickly by the time I've got to the top of the stairs I have to sit down and I have to recover."*

*([38], p9)*

Many participants experienced changes in taste, smell perceptions, and food behavior long after recovering from COVID-19 [31,33,40].

*"Just a small thing, a different sound, different look, or different taste, then I instantly loose the desire to eat, and then I give up."*

([33], p10)

Cognitive impairments and overall health decline critically impacted people with long COVID [38,40,41]. Participants often experienced difficulties in performing daily activities or doing things that they liked to do.

*"I can't cope with multiple inputs, like if I'm trying to reply to a message on my phone and one of my boys starts speaking to me or there's something else happening as well that just really fries my brain. I mean I used to be the kind of person that, like all women, multi tasking was a superpower. I was able to, do lots and lots of things, you know I'm [a doctor]; I would have one patient I'd be hearing lots about another patient coming I'd be remembering I'd be doing something else I'd be juggling lots and lots of things and now I can't keep multiple plates spinning I absolutely can't. I've got to focus on just one thing or I make massive mistakes and it's like I forget my intentions all the time.'"*

([41], p5)

Living with persistent symptoms, variations in the magnitude and patterns of health problems, uncertainty of recovery, and relapse of long COVID symptoms were major concerns reported by the participants [27–30,40]. These conditions made prolonged impacts on their lives as they couldn't reinitiate daily activities and had lesser control over their lives [38,40].

*"Sometimes it worries me a bit because there's people on there who've had these symptoms for 18 months, and I'm thinking oh my God, please no."*

([29], p7)

*"And I'm sure I've got a long way to go but so I'm in a much better place than I was but yes, I was really frightened, terrified and just thought I might die on a couple of occasions. . . maybe not "I'm going to die right now", but definitely "I'm never going to get better from this" kind of feeling."*

([27], p8)

Despite these challenges, many participants continued adapting to their health conditions and attempted to return to an altered lifestyle [32,35]. Learning about their health status and how to live with complex problems became a key component of their post-COVID experiences [38].

*". . . I feel there's an opportunity for change. I might reduce my hours going forward. It's difficult but I might try and balance my work-life balance a bit more and pace myself."*

([35], p5)

*". . .I do some things, but clearly not like before . . . I mean, do some things that before took me half a day, now it takes me four or five days"*

([32], p6)

## Synthesized finding 2: Psychosocial crises of long COVID

Long COVID patients reported experiencing psychosocial problems that were expressed in 18 findings (6 unequivocal and 12 credible) that were classified into 11 categories as summarized in Table 4. Mental health problems coexisted with physical problems impacting long-term wellbeing among many participants [30,34,35]. Several participants perceived loneliness and a sense of self-hatred that critically affected their mental health conditions [30,34].

> "I have noticed my anxiety has increased a lot. I know it's post-COVID symptoms but sometimes I doubt myself and think there is something else? it's very worrying however; I can manage my symptoms but it's more anxiety that creates more issues!"

([34], p7)

> "This is now of course affecting my mental health and I feel low in mood at times, and struggling with living alone but also hate requiring other people's help. I am often too tired to communicate anyway. I hate being dependent."

([30], p6)

Experiences of long COVID included changes in self-image, awareness, and identity among the affected individuals [27,38–41]. Their personal and professional identities were deeply challenged as their narratives of living with long COVID impacted how they felt about themselves. The psychological evolution of self-image and identity amidst long COVID was difficult for many participants.

**Table 4. Findings and categories within synthesized finding 2.**

| Synthesized finding 2 | Categories | Findings |
|---|---|---|
| **Psychosocial crises of long COVID** | Changed self-awareness and self-identity | Threat to individual identity (U) |
| | | Changes in self-identity (U) |
| | New self-image | Changes in self-image (C) |
| | | Self-doubt and loss of self-worth (U) |
| | Emotional disturbances | Emotional distress (C) |
| | Hopeless and fatalism | Setting deadlines (U) |
| | | Feeling horrified and uncertain (C) |
| | | Fatalistic attitude (C) |
| | Fear of infection | Fear of reinfection (C) |
| | | Fear of infecting other (C) |
| | Feeling guilty, ashamed, or stigmatized | Feeling of guilt at being ill (C) |
| | | Fear and stigma of infection (C) |
| | Financial challenges | Financial distress (U) |
| | Loneliness and social isolation | Loneliness and social isolation (U) |
| | | Loneliness and self-hatred (C) |
| | Psychological problems associated with chronic conditions | Mental health impacts on altered taste and smell (C) |
| | Chronic psychological stressors | Psychological problems (C) |
| | Stigma on mental health problems | Social stigma on mental health problems (U) |

*"I have found it very difficult to dissociate my doctor brain from my patient brain. I found it very difficult to. . . I'm a trainer as well, and I found it very difficult to dissociate my educators' brain from my patient brain so I've had that dynamic going on for several weeks. I said to him 'I hope I've handed over that locus of control, I'm putting trust in you, you're looking after me, I will go by your advice."*

*([39], p839)*

*"So, it's almost like everything's gone on a go slow. I feel that some days I wake up and I almost feel like I've kind of jumped forward 20 or 30 years and I feel like a little old lady instead of the active mum that I was six months ago."*

*([38], p10)*

Fear of being ill, reinfection, or infecting others affected the psychosocial wellbeing of long COVID patients. Intrapersonal fear of self-health amidst COVID-19 created mental distress, whereas the fear of infecting someone else impacted interpersonal communications. Moreover, fear of expressing mental distress reflected social stigma that affected sharing concerns with others.

*"Everyone is suspicious. Basically, everyone is afraid."*

*([32], p8)*

*"I think the stigma that was I mean around being depressed and having any mental conditions. I think people kind of judging for that. I've seen people do that. Especially in my friends' circle."*

*([40], p6)*

Financial distress was a key challenge affecting the mental wellbeing of many long COVID patients. Insufficient access to resources that could support recovery and economic wellbeing was experienced by many participants, which informed the need for socioeconomic support for the affected individuals.

*"I feel there is not much help for survivors in terms of emotional and financial support. One is left to figure out how best to manage on their own. . . because you do not have the financial support especially as a single parent you have to force and drag yourself to work in order to pay bills. Instead of concentrating on the healing process you are left with no choice but to work."*

*([34], p8)*

Emotional disturbances leading to anxiety and panic attacks affected many individuals. People struggled to set or meet deadlines and experienced uncertainty that affected their attitude to life.

*"This thing caused me anxiety and panic attacks, it was bad for me"*

*([31], p6)*

*"Truthfully, I was horrified. With the little bit of energy I had, I cried. I knew that from that point on, I nor anyone else had any idea what the next few days, weeks, months, or even years looked like for me."*

*([30], p6)*

## Synthesized finding 3: Slow recovery and difficult rehabilitation

Long COVID patients expressed their experiences of recovery from their health problems and concerns regarding rehabilitation. This finding was synthesized on 10 categories from 20 findings (8 unequivocal, 12 credible) as summarized in Table 5. Many of them explored physical, mental, and social activities that assisted the management of their symptoms and supported a more active and engaged lifestyle [27,35,38].

> "...facial exercises and massages for your lymph areas and everything and that's been proven to work with chronic fatigue, so for the last week, I've been trying to do that, you know with hot and cold complexes, compresses on your spine."
>
> ([38], p7)

> "So as much as I'm enjoying [walking the dog], it has the knock-on effect. But that is getting less and less, so the more I'm doing the better I'm feeling afterwards. I think [relapses are] all part of it, just got to get on with it and push myself a little bit harder and then hopefully I'll get better quicker. It doesn't put me off."
>
> ([35], p4)

Returning to work was a major challenge reported by many individuals with long COVID. Existing biopsychosocial problems affected their abilities to come back to work [34]. In addition, the workload in occupational settings appeared to be problematic given the slow pace of recovery from COVID-19 [38]. Such problems often created fear of returning to work or losing professional identity among the affected individuals [36]. A growing need for supportive

**Table 5. Findings and categories within synthesized finding 3.**

| Synthesized finding 3 | Categories | Findings |
|---|---|---|
| **Slow recovery and difficult rehabilitation** | Patient advocacy | Expert patients as changemakers (C) |
| | | Advocacy through online support groups (C) |
| | Balancing symptoms and performing activities | Balancing symptoms and physical and cognitive activity (C) |
| | | Exercise as a balancing act (C) |
| | Coping strategies for physical exercise | Developing personal strategies for activities (U) |
| | Coping with dietary practices | Dietary and nutritional adaptation (C) |
| | Returning to work | Inability to return to work (U) |
| | | Fear of coming back to work and loss of professional identity (U) |
| | | Physical and mental strain of work (U) |
| | | Work supporting recovery (U) |
| | | Lack of understanding from others (C) |
| | Hope and optimism | Hoping for recovery (C) |
| | | Use of mindliness (C) |
| | Symptoms management | Managing symptoms (U) |
| | | Self-management of symptoms (C) |
| | Rehabilitative and supportive care | Need for rehabilitation and supportive care (C) |
| | Alternative and complementary therapies | Preventive health behavior (C) |
| | | Use of complementary therapies (U) |
| | | Self-help (U) |
| | Psychological coping | Psychological adaptation (C) |

strategies in workplaces and empathetic understanding from colleagues was reported that could improve work-life balance [38,41].

> *"Can't go back to work as I still feel so unwell and my work (I am a teacher) are trying to push me into going back as I'm out of the 14-day infectious period and they are short staffed."*
>
> *([34], p8)*
>
> *"...my stock-in-trade is writing and using and testing outdoor equipment, so suddenly I can't... if you can't walk up a hill, you can't test a rucksack, you can't test boots, you can't use this stuff and if you can't use your brain to write stuff...effectively, it's rendered me incapable of working."*
>
> *([38], p10)*

Many individuals with long COVID reported several coping strategies during their recovery. Developing personalized coping approaches helped to overcome long COVID-related challenges [35]. Moreover, psychological coping facilitated better mental health and less psychological distress related to the symptoms. Many participants reported dietary and nutritional coping that supported their overall wellbeing [41]. In such approaches, many people emphasized the role of hope and mindfulness during recovery [33,36].

> *"I do the physical things that look after my mental health. So going outside and getting some fresh air looks after my mental health, and it in doing so helps the other symptoms, if that makes sense. So I focus on those a lot. So on a sunny day I'll go outside, because blue skies do me the world of good."*
>
> *([35], p5)*
>
> *"Psychologically, I feel better than before because I didn't use to appreciate some things, now I do, more. The sense changed a little, I want more to live, I'm happy, basically. I can't say anything else."*
>
> *([32], p9)*
>
> *"I do not want to be negative about it. It is going to be some long days."*
>
> *([33], p11)*

Self-help and the use of complementary therapies were reported by many participants who believed that such measures supported their recovery [30,39]. Many of them acknowledged the importance of preventive health behavior during recovery and rehabilitation [38]. Moreover, patients expressed their opinions on engaging them in healthcare decision-making and exploring potential opportunities for patient advocacy and empowerment [29,36].

> *"My GP tried to be supportive but has refused a referral to a 'long Covid clinic' as they say there is not one locally. I am taking multivitamins, folic acid, B-12, cod liver oil, Vit D, CoQ10 and turmeric religiously in an attempt to try and help myself."*
>
> *([30], p8)*
>
> *"I think the groups bringing it to light, that there is so many thousand people feeling this way means that someone's gonna have to step up and do something."*
>
> *([29], p6)*

## Synthesized finding 4: Digital resources and health information management

Thirteen findings (11 credible and two unequivocal) were classified into seven categories that informed a synthesized finding on digital resources for health and information management, as summarized in Table 6.

In several studies, many study participants expressed concerns regarding the availability and accessibility of personalized digital health support [29,37,39].

*"Having a virtual support group would be very helpful- a virtual patient group. A Facebook page for long covid can be helpful but it can also hinder, it can also send you off to different directions."*

*([37], p4)*

*"I thought it'd be good to hear from other medical systems as well, maybe they've got other ideas or different ways of tackling this."*

*([29], p7)*

While many participants found diverse perspectives and opinions in digital platforms [29], some participants reported quality issues about existing digital resources [39]. Also, online infotainment resources were used for gaming and entertainment by many participants [40].

*"I think it's a really good way of getting information out there, it just needs to be good quality information."*

*([39], p838)*

*"That (online game) has been an absolute lifesaver for me because it's kept my brain ticking over a bit, and given me, just a challenge, so I'm very fortunate with that. I'm sure other people*

**Table 6. Findings and categories within synthesized finding 4.**

| Synthesized finding 4 | Categories | Findings |
|---|---|---|
| **Digital resources and health information management** | Diverse opinions and perspectives in digital media | Diversity and global outreach in online support groups (C) |
| | Digital health resources | Need for digital health services and resources (C) |
| | | Quality of digital health resources (C) |
| | | Need for more accessible and personalized digital resources (C) |
| | | Need for interactive online support (C) |
| | Online infotainment | Online gaming and entertainment (C) |
| | Sharing health information | Access to health information (U) |
| | Health misinformation in social media | Trying to find answers (C) |
| | | Dealing with misinformation in social media (C) |
| | Need for information and guidance on recovery | Lack of clear and consistent physical activity-related advice (U) |
| | Knowledge development, dissemination and deployment | Need for more evidence (C) |
| | | Sharing experiences and knowledge (C) |
| | | Seeking knowledge in online groups (C) |

*have other things, but because I've got, naturally a competitive nature, with my golf, and I used to play tennis."*

*([40], p5)*

People with long COVID acknowledged the need for collaborative knowledge development, seeking evidence and experiences, and sharing the same [29,34,39]. Their continued attempts to find information on social media increased their vulnerability to health misinformation that emerged as an infodemic concern [27,40].

'... *Internet support groups, yeah on the Facebook groups that I'm on, I mean to be honest, I try not to read that group too much because it depresses me, makes me a bit anxious.*'

*([27], p6)*

*"So, at least I feel like they're trying to do something. There's a lot on there, which is negative. I try not to read that because . . . But that's like any social media isn't it, you just have to choose."*

*([40], p6)*

Many participants reported the need for specified and coordinated information management that may support their wellbeing through shared knowledge and informed decision-making [35,37].

*"Doctor should call us more and send information. We haven't had much from anyone else. That is what we lack, even if it is just a phone call every now and again."*

*([37], p4)*

## Synthesized finding 5: Changes in the perceived social support during long COVID

People with long COVID reported the changes in their social support, which were reported in 14 findings (4 unequivocal and 10 credible) that were classified into six categories (Table 7). Many participants wanted to share their concerns with their social networks but faced challenges in doing so [32,40]. Social exclusion and isolation were associated with declined social support that critically impacted the overall wellbeing of people with long COVID.

*"A lot of people don't understand long COVID, so when you explain to them, I'm still not feeling right 6 months down the line, a lot of people have said, I think you're just worrying too much. That's what I think my parents come back with. They keep saying to me, you worry too much, there's nothing wrong with you."*

*([40], p6)*

*"I want to be able to have laughter, I want to be able to go out and be the life and soul of the party, which I'm just not anymore, so I do grieve."*

*([40], p4)*

Many participants emphasized the roles of friends and family members. In addition, they offered good social support that enabled the long COVID patients to adapt to their new

Table 7. Findings and categories within synthesized finding 5.

| Synthesized finding 5 | Categories | Findings |
|---|---|---|
| **Changes in the perceived social support during long COVID** | Altered social life | Being unable to maintain pre-COVID normal social life (U) |
| | | Limited social activities (C) |
| | Adapted communications in friends and family | Coping with altered family communications (C) |
| | Changes in social support | decline social support over time (C) |
| | Online support groups | Finding support groups and communities of practice (U) |
| | | Online peer support (U) |
| | | Managing online support groups (C) |
| | | Psychological wellbeing among online group members (C) |
| | | Online social support (C) |
| | Social support from friends and family | Role of family and friends in healthier lifestyle (C) |
| | | Social connectedness and support (C) |
| | | Role of social networks to access care (C) |
| | | Receiving support and empathy from social network (U) |
| | Role of peer support | Social support from people with similar conditions (C) |

lifestyle, perform physical activities, stay mentally strong, and restore daily activities [33,38–40].

*"I started going out every day and trying to do a bit more of a walk at home where it was relatively safe, my mum was there, she would walk with me and look after me."*

*([33], p5)*

*"I've built up a little network of friends and I walk with friends so that it takes my mind off what I don't, what I'm so scared about it becoming too, I don't want to start internalising. So rather than going out on my own and thinking about stuff I go out with friends and then I can just talk and do the exercise without. . .and take my mind off the illness really."*

*([38], p8)*

People often joined social media forums and online groups that connected them to peers with similar experiences. Sharing their views and feelings revalidated their health and life concerns, which offered social support while navigating through uncertain conditions together [29,30,36,40]. It was difficult for many participants to receive such support from their families or HCPs [30,36]. In such scenarios, peer support through online platforms facilitated the exchange of information from many people that was not feasible in physical settings or networks.

*"[. . .] when it comes from people who have gone through the same thing, it feels a little bit more as in it's realistic and it will happen."*

*([29], p6)*

*"And the best support I got was from all the Facebook groups, believe it or not. That's where I found a lot of information, because everyone else was on a similar timeline to me. So, we were all going through the same symptoms, so I knew I wasn't going crazy."*

*([40], p6)*

*"[. . .] these have been a lifeline for so many people, because (.) when the medical services were failing, this was a beacon of light for people."*

*([29], p8)*

## Synthesized finding 6: Experiences with healthcare providers, services, and systems

A total of 30 findings (14 unequivocal and 16 credible) on the experiences with HCPs including GPs, health services, and health and social welfare systems were included in nine categories within this synthesized finding, as summarized in Table 8. Long COVID patients reported difficulties in finding adequate support from HCPs [29,30,39]. Many of them reported fear of being perceived negatively by their providers or not receiving the clinical

**Table 8. Findings and categories within synthesized finding 5.**

| Synthesized finding 6 | Categories | Findings |
|---|---|---|
| Experiences with healthcare providers, services, and systems | Access to diagnostic services | Need for thorough clinical assessment and diagnosis (U) |
| | | Access to testing and diagnostic services (C) |
| | Access to healthcare services | Difficulty accessing and navigating services (U) |
| | | Accessing care (C) |
| | | Experiences of accessing GP services for long COVID (C) |
| | | Lack of access to care (C) |
| | Experiences with healthcare providers | Finding the 'right' GP (U) |
| | | Adverse experiences with GP services (U) |
| | | Perceived role of clinical support (U) |
| | | Fear of being perceived negatively by HCP (C) |
| | | Lack of perceived support from and satisfaction with HCPs (C) |
| | | Lack of support from HCPs (C) |
| | | Inadequate support from HCPs (C) |
| | | Symptoms validation by HCPs (C) |
| | Need for multidisciplinary care | Need for multidisciplinary support (U) |
| | | Availability of multidisciplinary clinics (U) |
| | | Need for multidisciplinary care (U) |
| | Quality care | Concerns about quality and safety of care (C) |
| | | Satisfaction with specialized care (U) |
| | Referral care services | Delaying referral (U) |
| | | Need for multidisciplinary referral care (C) |
| | Health communication | Open therapeutic communications and relationships (U) |
| | | Supportive health communication (C) |
| | | Role of supportive communication with HCPs (C) |
| | | Empathetic communication with HCPs (U) |
| | Experiences with health systems and services | Unkind and uncompassionate system (C) |
| | | Health systems challenges for strengthening people-centered care (C) |
| | | Improve primary care services to support patients (U) |
| | | Navigating the healthcare system (C) |
| | Continuum of care | Lack of continued care (C) |

support they wanted. These challenges affected their overall experiences with HCPs working in diverse healthcare settings such as inpatients, outpatients, and community-based practices.

*"And my GP wasn't really very interested in it. I think at my kind of insistence she discussed with a medical consultant at the hospital, and the consultant said, "Well, that's normal for COVID. That's what people are experiencing, so there's no investigations needed", which to me didn't feel remotely reassuring."*

*([39], p837)*

*"[. . .] she said (mimicking voice) I think you probably do have long COVID. And that was like full stop. I kind of paused waiting for some—there was nothing."*

*([29], p5)*

During patient-provider interactions, many patients reported a need for open, supportive, and empathetic health communication that could address their concerns and facilitate health services utilization in their respective contexts [28,38,40,41].

*"I found it a struggle talking to my doctors, I felt like I wasn't being believed, like I was just, you know, making stuff up or bigging things up."*

*([38], p6)*

*"In the early days, my GP was fantastic . . . I sent him a letter to tell him what was going on in my life, all my symptoms and everything. And I said, I am so sorry about harassing you. And he phoned me up, and he said, keep harassing me, he said, if you don't tell me, I don't know."*

*([40], p5)*

People with long COVID reported having poor access to diagnostic, therapeutic, and referral services [34,37,39]. In many scenarios, accessing and navigating existing services were difficult, whereas delays in timely access to care affected the health and overall wellbeing. A critical need for access to multidisciplinary and continued care was perceived by many study participants both during hospitalization and even after being discharged from care facilities.

*"I would really like a thorough examination, (chest x-ray for example) to determine precisely what the problems are and how best they can be resolved."*

*([34], p8)*

*"I was sometimes asking for appointments every other day because we couldn't get in touch with the hospital. The doctors there wouldn't respond. . . It was literally a complete barrier."*

*([39], p837)*

*"What's bothering me at the moment is I'm not in any plan or regime. So, I'm not on any pharmacological medicine. I'm not on any physical rehabilitation. So, I'm not on any programme. So, I just feel like I'm ambling. I'm just in the ether . . . A year of my life has gone. I can't carry on."*

*([40], p5)*

Several studies reported concerns regarding the quality and safety of existing services for people with long COVID. At the systems level, many patients reported the need for strengthening healthcare systems and creating compassionate, supportive, and people-centered services across healthcare organizations that would be responsive to and responsible for providing humane care to the affected individuals [28,36,37,41].

*"They said ok we'll get someone to phone you. My GP called back and just said 'oh well it's probably anxiety'. He didn't seem to have any idea what it could be. I felt fobbed off. I said I'm worried—there are articles and news outlets that I've been reading and I want to know what's happening to me—people are having strokes, blood clots. I haven't been to hospital but I'm concerned I'm still getting these effects. He said 'oh you'll be fine you've only had it mildly'."*

*([28], p8)*

*"I know quite a few of the people at the health centre personally, and I think they're all really good people and really caring people but the experience from my end has not been one of care, particularly."*

*(A doctor as a long-COVID patient, [36], p62)*

## Discussion

This systematic review and meta-synthesis evaluated qualitative evidence on the experiences and perspectives of people with long COVID that may improve the knowledge base and facilitate informed practice. One of the major findings of this review was how people are living with chronic physical health problems weeks to months after COVID-19 infection. This qualitative evidence is consistent with emerging research on musculoskeletal, neurological, cardiorespiratory, and other systemic disorders among people with long COVID [10,12,13]. A meta-research among 47,910 COVID-19 patients found that 80% of them developed one or more long-term symptoms. The most commonly reported symptoms included fatigue, headache, attention disorder, and dyspnea, among other health problems [13]. This growing burden of physical health problems necessitates effective management using the best available evidence.

Another synthesized finding of this review informed a heavy burden of psychosocial challenges associated with long COVID that affected the overall wellbeing and quality of life. Several research syntheses suggest a high prevalence of mental health problems such as anxiety, depression, posttraumatic stress disorder, psychological distress, and sleep disorders during COVID-19. Many of these problems exist long after recovering from the primary infection with coronavirus, which informs a hidden burden of mental health crises across populations. Several studies demonstrated a high prevalence of mental health problems in different patient groups with long COVID, particularly those with preexisting psychological problems or persistent health problems, which require more clinical attention [42,43]. Mental healthcare must be prioritized for people with a history of COVID-19 who are highly vulnerable to such problems. Moreover, mental health services should be integrated within primary care and community-based healthcare that may enhance access to those services.

Patients' experiences with recovery from COVID-19 and rehabilitation inform prolonged periods of suffering long COVID, limited progress, and uncertainty of returning to normal health [32,37,38]. Individuals may follow common health advice such as exercise and physical therapies [7,35], whereas many people adopt self-management strategies such as consuming nutritional supplements or taking alternative therapies [31,33]. While such measures reflect their necessity to recover from long COVID symptoms, any therapy without adequate

evidence of effectiveness and safety may result in inadvertent consequences. Patients and their caregivers should be educated on such risks and emerging evidence-based self-management practices that may support their recovery. Moreover, the continuum of care should be ensured with proper planning, considering the long course of recovery. Furthermore, several occupational implications of recovering from COVID or returning to work are increasingly recognized [44–46], which inform the need to reform workplace policies and practices supporting workers' wellbeing if they have long COVID symptoms.

This review found a growing adoption of online tools and resources among people with long COVID. Digital platforms enabled rapid sharing of information in times of uncertainty and empowered individuals to connect virtually with others [23,28,34]. These resources offered various support to individuals and helped them to cope with their health and social problems. However, psychosocial distress due to overwhelming information, misinformation, or disinformation may have profound population health impacts [47,48].

Many scholars argue that COVID-19 appeared as an infodemic [49–51], where the widespread distribution of health misinformation has affected health awareness, decisions, and outcomes across physical and virtual communities. People with long COVID should be empowered with education, tools, and resources that may help them to check the accuracy of the information they receive in multiple media. Managing or regulating infodemics on social media platforms can be difficult as there are critical concerns such as privacy, autonomy, and other sociolegal issues [52,53]. These problems require multidisciplinary research and actions from relevant stakeholders to prevent infodemics that affect health and wellbeing in populations [53].

Another major finding of this review was the changes in social support that people with long COVID perceived in their respective social networks [28,31,37]. Interpersonal relationships in the era of social distancing may have critical implications on how individuals and communities supported each other during this pandemic [54,55]. Understanding such dynamics may inform post-COVID social support development and social work practices. It is necessary to educate people about the psychological, emotional, and social aspects of individuals living with long COVID. Given the magnitude of this pandemic, most individuals and communities might need social support and care during or after COVID-19, which should be considered a social health priority. An inclusive and caring social system should be envisaged that may facilitate context-specific and socio-culturally appropriate interventions that may enhance social support and future health outcomes.

Several studies reported the experiences of long-COVID patients with healthcare providers (HCPs), services, and systems [27,28,37]. Several challenges were identified, including inadequate information and communication from HCPs, lack of understanding and empathy, and poor availability and accessibility of healthcare services. Such challenges are consistent with previous research on health systems and services that suggest a lack of resilience of health and social systems that affect healthcare delivery globally [56–58]. Also, provider challenges such as limited resources, burnout, lack of timely evidence and guidance, and poor communication skills have affected healthcare experiences of many people. These challenges, alongside other problems experienced by long-COVID patients, must be addressed using multilevel recommendations that are summarized in Table 9.

The current evidence informs critical research gaps that require the attention of the scientific community. Most studies in this review are from the UK, which highlights a lack of research representation from other nations, particularly from the low- and middle-income countries. Differences in health and social contexts could potentially shape the experiences of people with long COVID in different countries, which could enrich the evidence base. Moreover, many studies recruited participants online given the use of telecommunication and

**Table 9. Multilevel recommendations for health promotion in people with long COVID.**

| Level of engagement | Recommendations |
| --- | --- |
| **Patients** | • Empowering patients with high-quality information and best available evidence on long COVID, associated problems, and resources to address the same<br>• Engaging patients in shared decision-making processes<br>• Encouraging exchange of views, experiences, and opinions on health problems and potential solutions<br>• Educating patients about evidence-based self-management practices<br>• Ensuring timely and continued access to chronic care services |
| **Social networks and family caregivers** | • Educating family caregivers and people within patients' social networks about long COVID and related social challenges<br>• Exploring barriers and facilitators of social networking and support-giving processes to support the caregivers<br>• Engaging friends and family caregivers in shared decision-making on health promotion among the affected people<br>• Caregiving-related resources and services should be enhanced so that informal caregivers can support the affected individuals |
| **HCPs** | • Ensuring proactive and compassionate health communication<br>• Examining diverse health and social problems that may inform healthcare delivery processes and improve subsequent outcomes<br>• Emphasizing people-centered value-based care<br>• Ensuring easy-to-understand navigation through points of care and supporting patients journey through health services<br>• Organizing shared decision-making sessions engaging all health and social providers, patients, and family caregivers<br>• Providing evidence-based personalized/precision care considering unique set of biopsychosocial problems of individuals |
| **Employers and community organizations** | • Revisiting institutional policies and programs to support long COVID patients and modify their workstyles ensuring optimal health and wellbeing<br>• Engaging medical and public health professionals in organizational health promotion considering the long-term consequences of COVID among internal and external stakeholders<br>• Increasing access to exercise, meditation, and non-pharmacological supportive care can be incorporated within organizational spaces for facilitating sustainable recovery and positive health<br>• Monitoring and evaluation of health alongside organizational processes and outcomes should be prioritized for developing healthy workforce and teams |
| **Media and communication platforms** | • Developing accessible and personalized media platforms or contents for long COVID patients and caregivers<br>• Ensuring the quality and accuracy of evidence shared across platforms<br>• Preventing potential infodemics in digital media through fact-checking tools, increasing mass awareness, and strengthening regulatory measures<br>• Engaging media experts, health communication professionals, psychologist, and other professionals to design safe and people-centered media contents<br>• Enabling individuals and communities to use media adhering to best practices that promote health and prevent unintended consequences |
| **Healthcare organizations and systems** | • Recognizing the growing burden and the complex nature of long COVID-related problems<br>• Emphasizing timely access to diagnostic, therapeutic, referral, and rehabilitative care for managing health problems<br>• Adopting integrated multilevel and transdisciplinary models of care supporting people with long COVID through collaborations among HCPs<br>• Addressing policy challenges that may affect access to care for people with long COVID<br>• Identifying and supporting marginalized or most affected individuals and communities in a population who might need additional care to recover from long COVID<br>• Supporting long-term care and holistic wellbeing through low-cost, community-based, and rehabilitative care services<br>• Prioritizing pro-patient health communication and extending telehealth services for improving long COVID care coverage<br>• Promoting participatory health services development engaging patients and family caregivers and addressing their concerns<br>• Mapping health services gaps and addressing the same for achieving health equity across populations |

preventive measures amidst COVID-19. However, such online-based recruitment may systematically exclude the voices of people with a digital divide from respective studies. Further, underrepresented communities such as people of color or gender minorities may experience an added burden of health and social inequities that may affect their journey through long COVID. Future research should consider such inclusion to promote research equity. Also, different qualitative methods that are not used in current studies could have informed how the experiences of the participants are shaped socially and culturally or how their views are or can be used for context-specific actions. These are a few areas of research development that may facilitate future research on this topic.

There are several limitations of this systematic qualitative review that should be considered while interpreting and using this review. First, we included peer-reviewed articles only. Thus, many articles that were available in preprint servers or other non-reviewed sources were excluded from this review that may potentially affect the overall body of synthesized evidence. Second, we used scholarly databases as the main data sources and explored additional sources through hand-searching. Still, it is possible to have published pieces that were available in print versions only that we could not access. Third, our choice of meta-aggregation is widely used in qualitative systematic reviews. However, there are other ways of qualitative meta-research that could have provided additional or different insights compared to the current approach.

## Conclusions

People with long COVID experience a wide range of biopsychosocial challenges that impact their health and wellbeing. Existing models of care should be strengthened considering the health and socioeconomic needs of the affected individuals and communities. Also, the growing adoption of online resources and changing dynamics of social support should be examined to improve the quality of life among people with long COVID. The current evidence base informs the need for transdisciplinary research and actions for addressing health disparities associated with long COVID. It is necessary to acknowledge the complex nature of long COVID and develop a shared vision engaging patients, HCPs, policymakers, social workers, media professionals, institutions, and other stakeholders to mitigate long COVID and associated crises globally.

## Supporting information

**S1 File. PRISMA-P (Preferred Reporting Items for Systematic review and Meta-Analysis Protocols) 2015 checklist for this systematic review.**
(DOCX)

**S2 File. Quality appraisal of the included studies in this systematic review.**
(DOCX)

## Author Contributions

**Conceptualization:** M. Mahbub Hossain.

**Data curation:** M. Mahbub Hossain, Jyoti Das, Farzana Rahman, Fazilatun Nesa.

**Formal analysis:** M. Mahbub Hossain, Jyoti Das, Farzana Rahman, Farah Faizah.

**Methodology:** M. Mahbub Hossain, Neetu Purohit, Gilbert Ramirez.

**Project administration:** Fazilatun Nesa, Puspita Hossain, A. M. Khairul Islam, Hoimonty Mazumder.

**Visualization:** A. M. Khairul Islam, Samia Tasnim.

**Writing – original draft:** M. Mahbub Hossain, Jyoti Das, Farzana Rahman.

**Writing – review & editing:** M. Mahbub Hossain, Fazilatun Nesa, Puspita Hossain, A. M. Khairul Islam, Samia Tasnim, Farah Faizah, Hoimonty Mazumder, Neetu Purohit, Gilbert Ramirez.

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
