## [Decision Letter · Decision Letter 0]

16 Jan 2023

PONE-D-22-26432Living with “long COVID”: A systematic review and meta-synthesis of qualitative evidencePLOS ONE

Dear Dr. Hossain,

Thank you for submitting your manuscript to PLOS ONE. After careful consideration, we feel that it has merit but does not fully meet PLOS ONE’s publication criteria as it currently stands. Therefore, we invite you to submit a revised version of the manuscript that addresses the points raised during the review process.

We look forward to receiving your revised manuscript.

Kind regards,

Federica Canzan

Academic Editor

PLOS ONE

and https://journals.plos.org/plosone/s/file?id=ba62/PLOSOne_formatting_sample_title_authors_affiliations.pdf.

2. PLOS requires an ORCID iD for the corresponding author in Editorial Manager on papers submitted after December 6th, 2016. Please ensure that you have an ORCID iD and that it is validated in Editorial Manager. To do this, go to ‘Update my Information’ (in the upper left-hand corner of the main menu), and click on the Fetch/Validate link next to the ORCID field. This will take you to the ORCID site and allow you to create a new iD or authenticate a pre-existing iD in Editorial Manager. Please see the following video for instructions on linking an ORCID iD to your Editorial Manager account: https://www.youtube.com/watch?v=_xcclfuvtxQ.

Reviewers' comments:

Reviewer's Responses to Questions

**Comments to the Author**

1. Is the manuscript technically sound, and do the data support the conclusions?

Reviewer #1: Yes

Reviewer #2: Yes

2. Has the statistical analysis been performed appropriately and rigorously? 

Reviewer #1: Yes

Reviewer #2: Yes

3. Have the authors made all data underlying the findings in their manuscript fully available?

Reviewer #1: Yes

Reviewer #2: Yes

4. Is the manuscript presented in an intelligible fashion and written in standard English?

Reviewer #1: Yes

Reviewer #2: Yes

5. Review Comments to the Author

Reviewer #1: This meta synthesis provides an overview of patients' experience of a very relevant health problem in the current health care scenario.

In my opinion, a more accurate and detailed definition of what is meant by long covid is needed.

Add a more detailed description of the health problems long covid patients had, specifying and listing what the organs/apparatus most affected are, and what the consequences are.

The order of presentation of findings can be improved, following an order of importance.

Reviewer #2: the Authors analyze the symptoms and after-effects of the "long-covid", certainly a current topic. The analysis performed on the databases and on the data published in the literature is rigorous, the concluding considerations are appropriate and allow us to indicate fixed points in the evaluation of these patients. he symptoms of post-covid are in fact often difficult to interpret, this analysis allows an objective evaluation of what has been highlighted by the published studies.

6. PLOS authors have the option to publish the peer review history of their article (what does this mean?). If published, this will include your full peer review and any attached files.

Reviewer #1: **Yes: **Gabriele Chini

Reviewer #2: No

---

## [Author Response · Author response to Decision Letter 0]

24 Jan 2023

Thank you for your valuable comments and suggestions. We have added our responses to the same in a separate file attached to this revised submission.

---

## [Editor Report · Decision Letter 1]

2 Feb 2023

Living with “long COVID”: A systematic review and meta-synthesis of qualitative evidence

PONE-D-22-26432R1

Dear Dr. Hossain,

We’re pleased to inform you that your manuscript has been judged scientifically suitable for publication and will be formally accepted for publication once it meets all outstanding technical requirements.

Kind regards,

Federica Canzan

Academic Editor

PLOS ONE
---

## [Editor Report · Acceptance letter]

7 Feb 2023

PONE-D-22-26432R1 

Living with “long COVID”: A systematic review and meta-synthesis of qualitative evidence 

Dear Dr. Hossain:

I'm pleased to inform you that your manuscript has been deemed suitable for publication in PLOS ONE. Congratulations! Your manuscript is now with our production department. 

Kind regards, 

on behalf of

Professor Federica Canzan 

Academic Editor

PLOS ONE